# Levels of heavy metals in soil and vegetables and associated health risks in Mojo area, Ethiopia

Hailu Reta Gebeyehu[¤][�], Leta Danno Bayissa[iD]*[�]

Department of Chemistry, College of Natural and Computational Sciences, Ambo University, Ambo, Ethiopia

he These authors contributed equally to this work.
¤ Current address: Ethiopian Institute of Agricultural Research, Addis Ababa, Ethiopia
* bayissa.leta@ambou.edu.et

**Data Availability Statement:** All relevant data are within the paper and its Supporting Information files.

**Funding:** The project was financially supported by Ethiopian Institute of Agricultural Research (EIAR).

## Abstract

Health implications to the population due to the consumption of contaminated vegetables has been a great concern all over the world. In this study, the levels of heavy metals (Cr, Cd, Zn, Fe, Pb, As, Mn, Cu, Hg, Ni and Co) in soil and commonly consumed vegetables from Mojo area in central Ethiopia have been determined using Inductively Coupled Plasma Optical Emission Spectrophotometer (ICP-OES) and possible health risks due to the consumptions of the vegetables have also been estimated. The levels of As, Pb, Cd, Zn, Cu, Hg and Co were exceeded the reference level in agricultural soil. Likewise, As, Pb, Cd, Cr and Hg levels exceeded the recommended values in vegetable samples with concentrations ranging from 1.93–5.73, 3.63–7.56, 0.56–1.56, 1.49–4.63 and 3.43–4.23 mg/kg, respectively. It was observed that leafy vegetable (cabbage) has accumulated heavy metals to greater extent compared with tomato. The estimated daily intake (EDI) of toxic metals due to the consumption of the vegetables were below the maximum tolerable daily intake (MTDI). However, the total health quotient (THQ), calculated based on EDI of the heavy metals were found > 1 for As and Hg due to tomato consumption and for As, Hg and Co due to cabbage consumption, suggesting significant health risk. The health index (HI) due to the intake of toxic metals from the consumption of both vegetables were much > 1, with HI values of 7.205 and 15.078 due to tomato and cabbage consumption, respectively. This clearly suggests the possible adverse health effect to adult population from the consumption of tomato and cabbage from the study area. The total cancer risk (TCR) analysis have also revealed the potential adverse cancer risk induced by As, Cd, Hg, and Ni from the consumption of both tomato and cabbage as their TCR values were above the threshold level. Based on the results of this study, there would be a significant health risk (both non-carcinogenic and carcinogenic) to the consumer associated with the consumption of cabbage and tomato being cultivated in Mojo area. Consequently, we recommend a strict regulatory control on the safety of vegetables originated from the study area.

The funder had no role in study design, data collection and analysis, decision to publish, or preparation of the manuscript.

**Competing interests:** The authors have declared that no competing interests exist.

## Introduction

Environmental pollution by the heavy metals and associated food safety is a major global concern now a days. These metals can pose a serious health implication to all living things in general and humans in particular if accumulated in elevated concentration above body requirements as described by Gupta and Gupta [1]. It is an understandable fact that the demand and consumption of vegetable is significantly increasing all around the world as it constitutes an important part of the human diet and nutrition. Kachenko and Singh [2,3] have explained that commercial and residential vegetable growing are often located in urban areas and are subjected to anthropogenic contamination from various sources including: urban and industrial wastes, mining and smelting and metallurgical industries. As a result, food safety issues and potential health risk are becoming a major public concern worldwide and make it as one of the most serious environmental concerns [4].

Heavy metals are one of a range of important types of contaminants that can be found on the surface and in the tissue of fresh vegetables as reported by Bigdeli and Seilsepour [4]. It has been reported that, rapid industrialization and urbanization have contributed to the elevated level of heavy metals in the urban environment in developing countries [5]. Contamination of soil with heavy metals is common and it can be a major source of metals to crops and finally may be a primary path of human exposure to these potentially toxic metals [6–8]. Several scientific reports have identified heavy metals as significant contaminants of the vegetables being grown in and around urban areas all over the world [4, 8, 9–15]. This is a clear indication that vegetables being grown in and around urban and suburb areas are significantly susceptible to being contaminated with elevated amounts of heavy metals.

In developing nations including Ethiopia, small and medium scale industries are expanding in a fastest rate now a days, and are mostly established in and around urban areas and along the River banks. These industries are working mainly on metal processing, beverages, textiles, chemicals, floriculture, paints, paper, pesticide, cement, plastic and tanneries [16]. It has been widely reported that, wastewater from these industries are often contains high concentrations of heavy metals, including Cadmium (Cd), Arsenic (As), Mercury (Hg), Copper (Cu) and lead (Pb) and posing serous environmental problems [4, 7, 9, 14, 17–28]. As it can be clearly understood from literature, these elements at concentrations exceeding the physiological demand of the plants, not only could administer toxic effect in them but also could enter food chains, get biomagnified and pose a potential threat to human health [29–32].

All over the world, especially in developing nations, there is a growing public concern over the potential accumulation of heavy metals in soil, water and plants owing to rapid industrial development [33–36]. For instance, Kamani and co-workers [37] have reported the contamination levels of street dust with heavy metals and associated health risk to the population in Tehran, Iran. Similarly, it has also become a major environmental concern in Ethiopia as well due to continuous and rapid industrialization and urbanization [38–40]. In our recent report [41], we have unveiled an alarmingly higher concentrations of various heavy metals being indiscriminately thrown out to the surrounding environment from electronic and electrical materials maintenance shops. These is a clear indication that, wastes from industries are instigating a wider environmental problems and health hazards globally and particularly in developing nations including Ethiopia as the safety precaution and environmental safety laws are not well documented. Therefore, the present study aimed at assessing the levels of selected heavy metals in selected vegetables and soil samples collected from Mojo area farmlands in central Rift Valley of Ethiopia, where the rivers used for irrigation are reportedly [38] highly contaminated with heavy metals being released from various industries in the area. To this regard we have also investigated the possible health risk associated with dietary exposure to

these potentially toxic metals by calculating the estimated daily intake (EDI), target hazard quotient (THQ), hazard index (HI) and target cancer risk (TCR) (for arsenic, lead, cadmium, chromium and nickel).

## Materials and methods

### Geographical locations of the study area

Mojo (also transliterated as Modjo) is a town in central Ethiopia, named after the nearby Mojo River and located at a distance of about 73 km south of Addis Ababa in East Shewa Zone, Lome woreda of Oromia Regional State. It has a latitude of 8˚35′N and longitude of 39˚6′E with an elevation between 1788 and 1825 meters above sea level. Mojo is the home for many industries including medium-sized leather and textile factories, plastic factory, edible oil; factories and many more. The area is highly vulnerable to pollution as the waste management practices among the industries situated in the area is non or very low and it has been witnessed that most of these factories directly release their effluents to mojo river. No protected or threatened species or locations were involved in this study. Likewise, no special permission was required to conduct this study. The geographical location of the study area is as given in Fig 1.

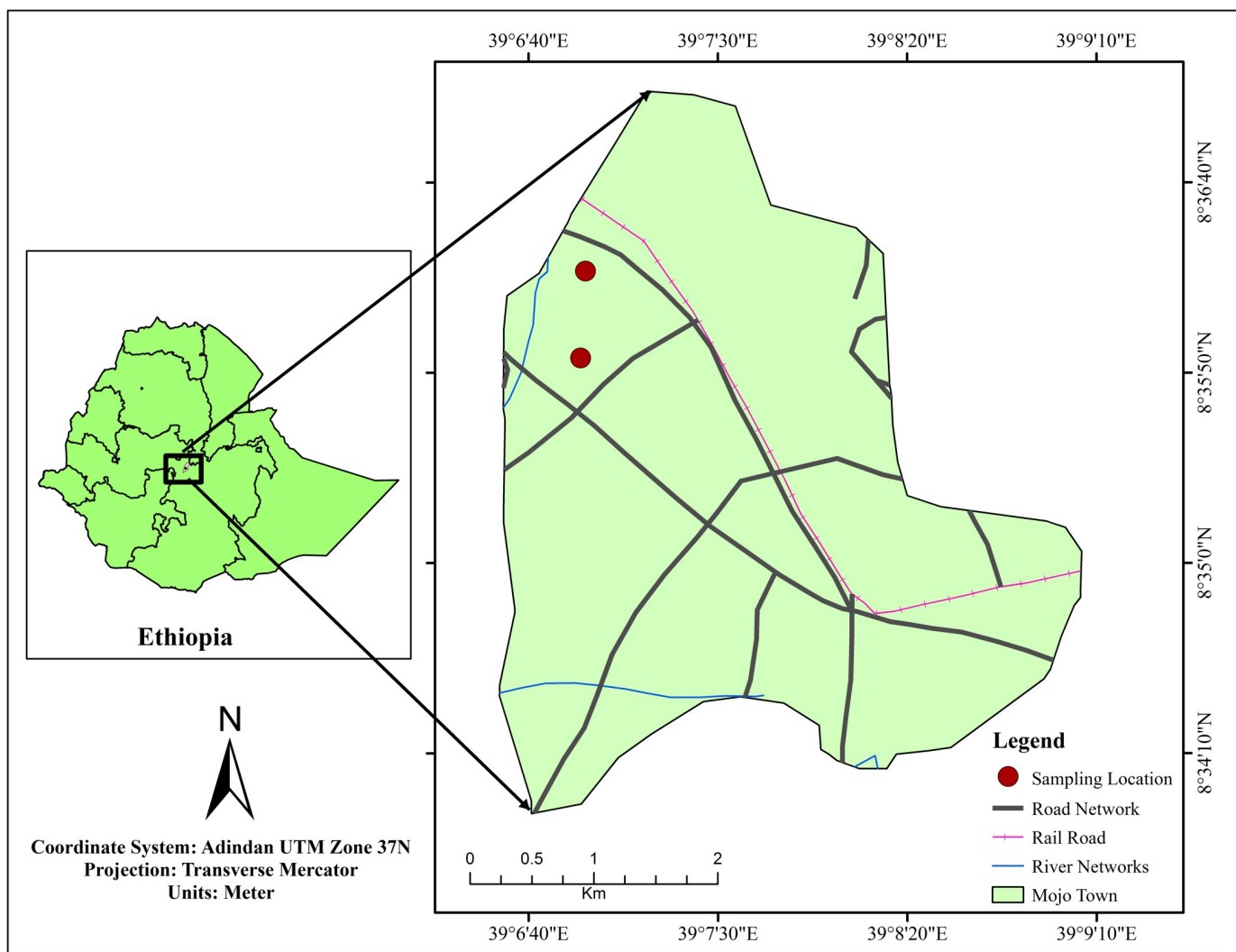

**Fig 1. Location map of the study area (drawn by ArcGIS 10.3 software).**

## Apparatuses and instruments

Properly cleaned and sterilized polyethylene bags have been employed for soil and vegetable samples collection. A Microprocessor based PH-EC-TDS Meter (Model: 1615, ESICO) was used for the determination of soil pH and conductivity. A Milestone Microwave (Model: STRAT D 134348, EVISA) was used for digestion of water, vegetable and soil samples, while Drying Oven (Model: DHG-9123A) was employed to dry the soil and vegetable samples. Analytical balance (Model E11140, Switzerland) was employed to weigh the processed samples, and measuring cylinders, pipette, and micropipette (Merck KGaA, Darmstadt, Germany) were used to measure different volumes of sample solutions, acid reagents and metal standard solutions. The digested samples were filtered with Whatman No. 42 filter paper and the digestion process were performed in a laboratory fume hood. The ICP-OES Spectro-Arcos (Model: ARCOS FHS12, USA) was used for the determination of target metals in water, vegetable and soil samples considered in this study.

## Chemicals and reagents

All reagents and chemicals used in this study were analytical grade, unless otherwise stated. Double distilled water was used for all preparation and dilution purposes of solutions throughout the experimental procedures. Chemicals such as $HNO_3$ (69%), ammonium acetate (≥98%), sodium acetate (≥99%), KCl (≥99%), HAc (≥99%), $MgCl_2$ (≥99%), $NH_2OH.HCl$ (98%), $H_2SO_4$ (98%) and $H_2O_2$ (30%) and HCl (37%) (all from Sigma Aldrich, USA) were used during sample digestion procedures. Stock standard solutions of 1000 ppm were prepared from their corresponding salts for the selected heavy metals (Cu, Zn, As, Cr, Fe, Mn, Ni, Pb, Cd, Hg and Co). Standard buffer solutions of pH = 4, 7 and 9 (from Macron Fine Chemicals™) were used for pH meter calibration and KCl (from Sigma Aldrich, USA) was used for conductivity meter calibration.

## Vegetables sample collection and preparation

The edible part of vegetable samples (cabbage (*Brassica oleracea*) and tomato (*Lycopersicon esculentum Miller*)) were collected into a precleaned and sterilized separate polyethylene bags. The vegetable samples were directly collected from privately owned farmlands with the consent of the owners (farmers). About 1 Kg of each (cabbage and tomato) were separately collected from five randomly selected subsampling sites and pooled together to form a composite sample. The bruised or rotten portions were manually removed and the remaining samples were carefully packed and immediately transported to the Agricultural and Nutritional Research Laboratory, Addis Ababa, for further processing and analysis. The vegetable samples were washed with tap water and distilled water in laboratory to remove adsorbed dust and particulate matters and then cut and chopped into small pieces using plastic knife in order to facilitate drying. Subsequently, the samples were air-dried for five days and further dried in hot air oven at 50–60°C for 24 hrs., to remove moisture and maintain constant mass. The dried samples were grounded into powder using acid washed laboratory mortar and pestle and then sieved using 2 mm mesh size sieve. The sieved samples were finally stored in polyethylene bags and kept in desiccators until digestion and analysis.

## Soil sample collection and preparation

Soil samples (about 1 Kg) were collected into a clean polyethylene bags from the same sites where the vegetable samples were collected (for each vegetable type separately) with the consent of the farmers at 0–20 cm depth using a steeliness steel auger and pooled together to form

composite sample. The collected, carefully packed and labeled soil samples were then transported to the Agricultural and Nutritional Research Laboratory for pretreatment and analysis. In laboratory, the soil samples were air dried in a dry and dust free place at room temperature (25˚C) for 5 days, followed by an oven dry until constant weights were attained. The samples were then ground with a mortar and pestle to pass through a 2 mm sieve and homogenized. The dried, sieved, and homogenized soil samples were finally stored in polyethylene bags and kept in desiccators until digestion and analysis.

## Optimization of digestion procedures for soil and vegetable samples

To select an optimum condition for digestion procedure, parameters like digestion time, reagent volume and digestion temperature were optimized by varying one parameter at a time while keeping the others constant. The optimum conditions were selected based on clarity of digests, minimum reagent volume consumption, minimum digestion time, simplicity and minimum and optimal temperature required to complete digestion of samples. Accordingly, a representative sample for both soil and vegetable were taken separately and digestion procedures were followed under controlled condition by varying one parameter while keeping others constant. From the optimization procedures, an acid volume of 9 mL of $HNO_3$ and 3 mL HCl at digestion time of 45 minutes, and pressure and digestion temperature of 80W and 180˚C, respectively, were found to be the optimal condition to digest 0.5g of each soil and vegetable samples (S1 Table).

**Digestion procedures for vegetable samples.** A 0.5 g of homogenized and powdered vegetable sample (each tomato and cabbage separately) was placed in microwave digestion vessel to which 9 mL of 10M $HNO_3$ and 3 mL 10 M HCL were added. The vessels were tightly capped and placed in the microwave digestion system and digestion take place at 180˚C for 45 mins until a clear solution was obtained. After digestion was completed, the clear and colorless solution obtained was filtered out through Whatman No. 42 filter paper into 50 mL volumetric flask and its volume was adjusted by 2% $HNO_3$. All the solutions prepared were then immediately assayed for heavy metals using inductively coupled plasma optical emission spectrometry (ICP-OES). Each vegetable sample was digested and analyzed in triplicate and the data reported is as mean ± SD. The blank solutions were prepared by following similar procedure as per the optimum conditions established and consequently analyzed.

**Digestion procedures for soil samples.** A 0.5 g of dried and homogenized soil samples were similarly transferred in to microwave digestion vessel in triplicate. In each of this vessel, 9 mL of 10 M $HNO_3$ and 3 mL 10 M HCl were added and the samples were digested at 180˚C for 45 mins. The clear solutions obtained were then filtered out through Whatman No. 42 filter paper to a 50 mL volumetric flask and finally diluted to the mark with 2% $HNO_3$ following procedure reported by [42] with slight modification as per the optimized conditions. The heavy metals of interest were then assayed by ICP-OES in triplicate and the data reported is as mean ± SD.

**Heavy metals analysis of the samples.** The concentrations of Cr, Cd, Zn, Fe, Pb, As, Mn, Cu, Ni, Co and Hg in the soil and vegetable samples were determined by using ICP-OES after properly calibrating the instrument using calibration blank and five working calibration standard solutions of each metals to analyzed. All the calibration procedures were evaluated based on their corresponding correlation coefficients ($r^2$) of the calibration curves which were found to be $\geq$ 0.998. In addition, the instrument's parameters such as plasma power, pump speed, coolant flow, Nebulizer flow and etc. were optimized for maximum signal intensity of the instrument based on the instrument's manual provided by the manufacturer (S2 Table).

**Bioconcentration factor (BCF).** It has been defined that bioconcentration factor is the ratio of heavy metal concentration in edible part of the plant to heavy metal concentration in soil sample [43,44]. Accordingly, heavy metal transfer from soil to plant was calculated as by the formula used by Kachenko and Singh, [3] and given in Eq (1).

$$BCF = \frac{C_{plant}}{C_{soil}} \qquad (1)$$

where $C_{plant}$ is heavy metal content in edible part of plant and $C_{soil}$ is heavy metal content in respective soil. The value of BCF greater than 1 indicates that the plant is a potential accumulator of for the metal being considered for analysis.

## Health risk assessments: Estimated daily intake (EDI), target hazard quotient (THQ), hazard index (HI) and target cancer risk (TCR)

**Estimated daily intake (EDI).** The estimated daily intake of the metals considered in this study were determined based on their mean concentration in each cabbage and tomato and the estimated daily consumption of the vegetables in gram. The EDI value of each metal of interest was determined by the formula used by [45] with slight modification as presented in Eq (2).

$$EDI = \frac{E_f \; x \; E_D \; x \; F_{IR} \; x \; C_M \; x \; C_f}{B_W \; x \; T_A} \; x \; 0.001 \qquad (2)$$

where $E_f$ is exposure frequency (365 day/year); $E_D$ is the exposure duration (65 years), equivalent to average life time [40]; $F_{IR}$ is the average food (vegetable) consumption (240 g/person/day), which were obtained from the World Health Report [46] for low fruit and vegetable intake; $C_M$ is metal concentration (mg/kg dry weight); $C_f$ is concentration conversion factor for fresh vegetable weight to dry weight (which is 0.085) [43,47,48]; $B_W$ is reference body weight for an adult, which is 70 kg [40]; $T_A$ is the average exposure time (65yrs x 365 days) and 0.001 is unit conversion factor. The overall data employed for the calculation of EDI is as compiled in Table 1.

**Target hazard quotient (THQ).** To assess non-carcinogenic human health risk from the consumption of vegetables contaminated by heavy metals, the target hazard quotient (THQ) values were estimated. The THQ values of the local population due to the consumption of contaminated vegetables were calculated using Eq (3) as described by Chen et al., [45], Khan et al., [49], Zheng et al., [50] and Ezemonye et al., [51].

$$THQ = \frac{EDI}{RfD} \qquad (3)$$

Where EDI is the estimated daily metal intake of the population in mg/day/kg body weight and RfD is the oral reference dose (mg/kg/day) values for each metals of interest and as listed in Table 1. If the value of THQ is $< 1$, it is generally presumed to be safe for the risk of noncarcinogenic effects and if it is $> 1$, it is supposed that there is a chance of noncarcinogenic effects with an increasing probability as the value upsurges [45,52].

**Hazard Index (HI).** It has been documented that the individual health risks of the analysed heavy metals in the same vegetable are accumulative and that is expressed as hazard index (HI) [43,45, 50–56]. Accordingly, the HI of target metals considered in this study are calculated using Eq (4) [52,53].

$$HI = \sum_{n=1}^{i} THQ_n; \; i = 1, 2, 3, \ldots, n \qquad (4)$$

**Table 1. Parameters and variables used in the calculation of EDI, THQ and TCR.**

| Parameters | | Vegetable Types | | References |
|---|---|---|---|---|
| | | Tomato | Cabbage | |
| $E_f$ (days) | | 365 | 365 | - |
| $E_D$ (years) | | 65 | 65 | [40] |
| $F_{IR}$ (g/day) | | 240 | 240 | [46] |
| $C_M$ (mg/kg dry weight) | | Table 4 | Table 4 | This study |
| $C_f$ | | 0.085 | 0.085 | [43,47,48] |
| $B_W$ (kg) | | 70 | 70 | [40] |
| $T_A$ (days) | | 23725 | 23725 | - |
| Oral reference dose (RfD) (mg/kg/day) | As | 0.0003 | 0.0003 | [52] |
| | Pb | 0.0035 | 0.0035 | [11] |
| | Cd | 0.001 | 0.001 | [52] |
| | Zn | 0.3 | 0.3 | [57] |
| | Cu | 0.04 | 0.04 | [57] |
| | Fe | 0.7 | 0.7 | [57] |
| | Mn | 0.14 | 0.14 | [57] |
| | Cr | 0.003 | 0.003 | [11] |
| | Hg | 0.0003 | 0.0003 | [11] |
| | Ni | 0.02 | 0.02 | [57] |
| | Co | 0.0003 | 0.0003 | [57] |
| Oral cancer slope factor ($CPS_o$) (mg/kg/day)$^{-1}$ | As | 1.5 | 1.5 | [52] |
| | Pb | 0.0085 | 0.0085 | [56] |
| | Cd | 0.38 | 0.38 | [58] |
| | Cr | 0.5 | 0.5 | [59] |
| | Ni | 1.7 | 1.7 | [57] |

where HI is the sum of various metals hazards. If the HI value became < 1.0, there is no apparent health impact due to the metals considered. However, an HI value of > 1.0 indicates potential health impact implication. A serious chronic health impact has been suggested for HI > 10.0 [52,53].

**The target cancer risk (TCR).** The cancer risk (CR) posed to human health due to the ingestion of individual possibly carcinogenic metals was estimated using Eq (5) as described by Sharma et al. [44]. Then, the target cancer risk (TCR) resulting from heavy metals (As, Pb, Cd, Cr and Ni) ingestion, which may promote carcinogenic effect depending on the exposure dose, were calculated using Eq (6) as described by Kamunda et al. [56].

$$CR = EDI \times CPSo \qquad (5)$$

$$TCR = \sum_{(n=1)}^{i} CR; \; i = 1, 2, 3, \ldots, n \qquad (6)$$

where *CR* represents cancer risk over lifetime by individual heavy metal ingestion, *EDI* is the estimated daily metal intake of the population in mg/day/kg body weight, *CPSo* is the oral cancer slope factor in (mg/kg/day)$^{-1}$ and n is the number of heavy metals considered for cancer risk calculation. The *CPSo* values for As, Pb, Cd, Cr and Ni are given in Table 1, while we couldn't find the corresponding value for Hg and hence its cancer risk was not calculated. It has been pointed out that the slope factor converts the estimated daily intake of the metal averaged over a lifetime of the exposure directly to incremental risk of an individual developing cancer [56].

**Statistical analysis.** The data obtained were subjected to ANOVA to investigate the effect of sample origin on the concentration of heavy metals. As the level of heavy metal contamination might vary with sample site, one-way ANOVA was used to test the existence of significant difference between the means. Besides, correlation coefficients were calculated to investigate the association between the target metals. In all statistical analyses, confidence level was held at 95% (unless otherwise indicated) and the statistical calculations were made using either Origin 2018 or SPSS software (Version 21).

# Results and discussion

## Physicochemical properties of soil samples

The data of physicochemical parameters analyzed for soil samples under tomato and cabbage cultivation from Mojo area farmland is as summarized in Table 2. The analysis of the soil samples has revealed that, both soil samples under tomato and cabbage cultivation have been found to have a clay soil texture in nature with the percentage clay, silt and sand compositions varying in the range of 42.78–43.8, 23.75–24.12 and 27.5–28.10%, respectively.

The pH of the soil samples analyzed were ranged from 8.21 to 8.31 showing that the soil in the area covered in this study is slightly alkaline in nature. The pH values obtained in this study were found to be slightly higher than values reported by [17] for soil samples irrigated with different water sources. The electrical conductivity (EC) obtained in this study have ranged from 1056.66 μS/cm for soil samples under tomato cultivation to 1062.18 μS/cm for soil samples under cabbage cultivation, showing no significant difference statistically at $p < 0.05$. The data we have obtained from this study is much higher than values reported by Alghobar and Suresha [17], which have ranged between 172–297 μS/cm. Mekki and Sayadi [60] on the other hand have reported a much higher EC values (which ranges from 3800–4050 S/cm) for soil samples saturated with phosphate processing wastewater. The higher levels of EC obtained in the soil samples correlates with the soil texture we have obtained. It has been demonstrated that soil with clay texture is expected to have higher EC which correlates strongly to soil particle size [61]. This in general suggests that the soil in the study area is loaded with mineral contents from the wastewater being used for irrigation.

The percentage organic carbon (% OC) of the soil samples considered in this study were found to be 1.22 and 1.18% for soil samples under tomato and cabbage cultivation,

**Table 2. Selected physicochemical properties of soils samples from farmlands around Mojo area in central Ethiopia.**

| Physicochemical parameters* | | Under tomato cultivation | Under cabbage cultivation |
|---|---|---|---|
| pH (1:2.5) | | 8.31±0.01[a] | 8.29±0.02[a] |
| Electrical Conductivity (μS/cm) | | 1056.66±1.52[a] | 1062.18±2.01[a] |
| Organic Carbon (%) | | 1.22±0.01[a] | 1.18±0.02 [a] |
| Organic matter (%) | | 2.10±0.0.02[a] | 2.13±0.01[a] |
| Moisture content (%) | | 29.03±0.01[a] | 28.79±0.11[a] |
| Cation exchange capacity (cmol (+)/kg) | | 42.78±0.37[a] | 43.08±0.52[a] |
| Soil Texture | % clay | 48.75±0.01[a] | 46.89±0.05[a] |
| | % silt | 23.75±0.25[a] | 24.12±0.34[a] |
| | % sand | 27.5±0.25[a] | 28.10±0.19[a] |
| Texture Class | | **Clay** | **Clay** |

*Mean values in the same row with the same small letter are not statistically different at $p < 0.05$.

respectively. The values are not statistically differed significantly from each other at 95% probability level (p < 0.05) in respective of the type of vegetables being grown on it. The % OM obtained in soil samples from this study is found to be very comparable with the data (1.18–3.29%) reported by Sharma et al., [44] data reported by, however, quite lower than the minimum content of 3.4% as mentioned by Plunkett [62]. In contrary to data we have obtained, Balkhair [35] have reported a 44.9% OM content in soil samples from western region of Saudi Arabia. The relatively lower % OM obtained in this study could be attributed to the excessive cultivation and soil erosion in the area. The percentage moisture content (% MC) of the soil samples analyzed in this study were found to be 29.03 and 28.79% for soil samples under tomato and cabbage cultivation, respectively, showing no significant difference at p < 0.05.

The cation exchange capacity (CEC) of the soil samples under tomato and cabbage cultivation were found to be 42.78 and 43.08 cmol (+) /kg, respectively. From literature it can be understood that the CEC of a soil generally increases with soil pH due to the greater negative charge that develops on organic matter and clay minerals [63]. It has also been indicated that CEC gives an insight into the fertility and nutrient retention capacity of soil [64]. The high CEC values we have obtained for the soil samples analyzed is a clear indication that the clay texture of the soil texture, particularly in combination with organic matter, possess a number of electrically charged sites, which can attract and hold oppositely charged ions as explained by Mukhopadhyay et al. [64].

## Optimization of the sample digestion procedures for metal analysis

Prior to sample extraction for the evaluation the levels of heavy metals sought in soil samples considered in this study, the sample digestion procedures have been optimized to attain an optimum condition for the digestion procedures. The optimization procedures were based on parameters including digestion time, reagent volume, optimal pressure and digestion temperature required. Accordingly, various experimental trials have been tested and an optimum condition which require an acid mixture of 9 mL conc. $HNO_3$ and 3 mL conc. HCl, digestion time of 45 minutes and optimal digestion temperature of 180˚C were found to be the optimal condition to digest 0.5g of the soil sample (S1 Table). The optimum condition achieved were selected based on clarity of digests, minimum reagent volume consumption, minimum digestion time and minimum temperature applied for complete digestion of the samples. The same procedures and digestion conditions have been followed for the digestion of 0.5 g of each of the vegetable samples after proper method validation as indicated under section 3.3.

## Method validation

**Method detection limit (MDL) and Limit of quantification (LOQ).** The limit of quantification (LOQ) and method detection limits (MDL) for all the metals considered in this study have been calculated from the response of seven replicates of the calibration reagent blank using a standard formula LOQ = 3 × SD and MDL = 10×SD and the data are presented in S3 Table. The data from the investigation have revealed that the values for MDL have ranged from 0.0002 to 0.0008 mg/L, while the corresponding LOQ value have ranged from 0.0006 to 0.001 mg/L. This is a clear indication that the instrument used was in good sensitivity for the analysis. The instrumental detection limits (IDL) given in S3 Table are as obtained from the instrument's operation manual.

**Precision and accuracy.** The precision and accuracy of the method employed in the investigation of heavy metals in both soil and vegetable samples were validated by matrix spike recovery analysis method and the data obtained were as given in S4–S6 Tables. As can be seen from the data, the recovery values obtained was ranged from 90 to 117.46% and the percentage

relative standard deviations (% RSD) were happened to be < 11.6% for all the samples. The matrix spike recovery obtained in this study falls within the acceptable range of 80–120% for a good recovery study [41]. The high percentage recovery obtained from the study validates the accuracy of the method and its reliability for the analysis of metal concentration in both soil and vegetable samples considered for analysis in this study. The lower %RSD values (< 15%) obtained indicated that the method proposed for this study is precise enough for the analysis of heavy metals.

## Levels of heavy metals in soils and vegetable samples

**Levels of heavy metals in soil samples.**   The levels of heavy metals in soil samples from Mojo area farmlands in central Ethiopia have been assayed and the data obtained is presented in Table 3. All the soil samples considered in this study were found positive for all the heavy metals analyzed. The mean concentration arsenic (As) were found to be 21.00 mg/kg in soil samples under tomato cultivation and 30.73 mg/kg in cabbage growing soil samples. The levels of arsenic in both soil samples analyzed were found to be greater than the safe limit of 20 mg/kg set by European Union, 15 mg/kg in paddy soil set by Japan [65] and 14 mg/kg reference value agricultural soil as reported by Brown [66]. Das and coworkers [67] have reported an arsenic concentration ranged from 7.31 to 27.28 mg/kg dry weight in soil samples from Bangladesh. However, a relatively much higher concentration of (51.52 mg/kg) arsenic level in surface soil from West Bengal, India have been reported [68]. Arsenic (As) has long been regarded as environmental contaminant though its use is still continued [69,70] and can be released to the environment via both natural (biogeochemical) and anthropogenic activities [71]. This can be witnessed by the high concentration of arsenic found in the soil samples we have analyzed in this study. The high levels of arsenic obtained could be attributed to the release of the metal and/or its compounds from the industries situated in the area. The analysis of variance for arsenic contents in soil samples under tomato and cabbage cultivation have showed the absence of significant difference at 95% probability level (p < 0.05).

The mean levels of lead (Pb) in the soil samples were found to be 37.93 and 35.80 mg/kg, respectively, for soil on which tomato and cabbage grown. The levels of Pd obtained in both soil samples in this study were found to be more than 3 times higher than the limit value for Pb (10 mg/kg) as cited by Sharma et al. [44], but much lower than Indian standard (250–500 mg/kg) as provided by Alghobar and Suresha, [17] in soil. Cadmium (Cd) on the other have values ranged from 4.76 mg/kg in soil samples under cabbage cultivation and 5.30 mg/kg for under tomato cultivation. The Cd concentrations we have obtained in this study were found to be much higher than the values reported by Sharma et al. [44] which was 0.79–1.73 mg/kg and also much higher than the limit value reported by Chang et al. [11]. Zinc (Zn) and copper (Cu) were also found in higher concentration in both soil samples collected from sampling locations with concentration range of 93.66 to 98.86 mg/kg for Zn and 25.50 to 25.96 mg/kg for Cu. The levels of Zn obtained in this study were found to be much higher than values (26.52–37.21 mg/kg) reported and limit value (50 mg/kg) cited by Sharma et al. [44]. Likewise, the level of Cu obtained in this study were also found to be higher than soil reference value (20 mg/kg) reported by Sharma and co-workers [44].

As can be seen from Table 3, iron (Fe) were found in the soil samples analysed with concentration of 41410 mg/kg in soil sample under cabbage cultivation and 46426.67 mg/kg in soil sample under tomato cultivation farmlands. The result of our investigation clearly indicated that the farmland soils from the study areas considered in this study are enriched with elevated concentrations of Fe, but our finding is less than the value (80000 mg/kg) reported in soil [72]. However, compared with values we have obtained, lower values of iron (11.3 to 62.2 mg/kg)

**Table 3. Levels of heavy metals (mg/kg) in soil samples collected from Mojo area farmlands in central Ethiopia.**

| Metals | Levels of heavy metals (mg/kg) in soil samples | | Reference values (mg/kg) |
|---|---|---|---|
| | Under Tomato Cultivation | Under Cabbage Cultivation | |
| As | 24.50±0.60 | 24.06±0.05 | 14[a] |
| Pb | 37.93±0.0 | 35.80±0.17 | 10[b] |
| Cd | 5.30±0.3 | 4.76±0.15 | ≤ 0.3[c] |
| Zn | 98.86±1.45 | 93.66±1.92 | 50[b] |
| Cu | 25.96±0.3 | 25.50±0.62 | 20[b] |
| Fe | 46426.67±141.80 | 41410.00±191.57 | - |
| Mn | 1763.33±47.25 | 1696.67±15.27 | 2000[d] |
| Cr | 36.23±0.4 | 35.93±0.30 | 100[b] |
| Hg | 6.26±0.40 | 7.30±0.43 | ≤ 0.3[c] |
| Ni | 35.58±0.56 | 30.50±0.81 | 50[d] |
| Co | 15.13±0.30 | 14.93±0.25 | 8[b] |

[a][66]

[b][44]

[c][11]

[d][54]

have also been reported by Rattan et al. [43]. The levels of manganese (Mn) in the soil samples studied were found to be 1763.33 and 1696.67 mg/kg for soil under tomato and cabbage cultivation, respectively. The levels of Mn obtained in soil samples of current study were found slightly lower than the reference value of 2000 mg/kg reported by Mahmood and Malik [54].

The levels of Cr, Hg, Ni and Co in the soil sample under tomato cultivation were found to be 36.23, 6.26, 35.58 and 15.13 mg/kg, respectively. The corresponding value in soil samples under cabbage cultivation were 35.93, 7.30, 30.50 and 14.93 mg/kg, respectively. The levels of Cr and Ni obtained in soil samples considered in this study are found to be less than the

**Table 4. Levels of heavy metals (mg/kg dry weight) in tomato and cabbage samples cultivated around Mojo area farmlands in central Ethiopia.**

| Metals | Levels of metals (mg/kg dry weight) | | Allowable concentrations (mg/kg) |
|---|---|---|---|
| | Tomato | Cabbage | |
| As | 1.93±0.50 | 5.73±0.37 | 0.1[a] |
| Pb | 3.63±0.11 | 7.56±0.23 | 0.1–0.3[ab] |
| Cd | 0.56±0.05 | 1.56±0.05 | 0.05–0.2[ab] |
| Zn | 24.50±0.43 | 23.53±0.11 | 50[c] |
| Cu | 16.27±0.40 | 9.42±0.15 | 10–40[ab] |
| Fe | 85.10±0.17 | 490.46±3.18 | - |
| Mn | 27.20±0.34 | 302.23±3.10 | 500[c] |
| Cr | 1.49±0.01 | 4.63±0.20 | 1–2.3[ac] |
| Hg | 3.43±0.05 | 4.23±0.28 | 0.01–0.3[bd] |
| Ni | 1.86±0.05 | 4.13±0.20 | 10[a] |
| Co | 0.63±0.05 | 1.86±0.05 | 50[c] |

[a] [55]

[b] [53]

[c] European union standards [54]

[d] Dutch target value [73]

recommended value of 100 mg/kg for Cr and 50 mg/kg for Ni as reported by Sharma et al. [44] and Mahmood and Malik, [54]. However, the level of mercury (Hg) obtained in the soil samples investigated was found to be much higher than the recommended value (0.3 mg/kg) for agricultural soil as reported by Chang et al. [11]. Likewise, the levels of cobalt (Co) obtained in soil samples under this study were found to be greater than the reference value of 8 mg/kg reported by Sharma et al. [44]. The result of our study in general have revealed that, the soil in the study area is clearly contaminated with highly toxic metals including As, Pb, Cd, Zn, Hg and Co, as their levels have significantly exceeded the reference values for agricultural soil.

**Levels of heavy metals in vegetable samples.** The levels of heavy metals in vegetable samples (cabbage and tomato) cultivated around Mojo area farmlands have been investigated and the result is presented in Table 4. The results of the investigation have shown that the mean levels of arsenic were 1.93 and 5.73 mg/kg (dry weight) in tomato and cabbage samples analyzed, respectively. The mean levels of arsenic in the vegetable samples were found to be much greater than the recommended value of 0.1 mg/kg as reported by Shaheen et al. [55]. For instance, the arsenic level obtained in cabbage samples were much higher than the values obtained in tomato sample. This is a clear indication that leafy vegetables accumulate arsenic in significant amount compared with fruity vegetable (tomato in this particular study).

The levels of lead and cadmium in tomato sample considered in this study were found to be 3.63 and 0.56 mg/kg, respectively, while the corresponding values in cabbage were 7.56 and 1.56 mg/kg for Pb and cd, respectively. The values of both Pb and Cd metals obtained have shown statistically significant difference in cabbage and tomato vegetable types at 95% probability levels ($P < 0.05$). It can be clearly observed here is that leafy vegetable (cabbage) happened to accumulated more toxic metals than the fruity vegetables (tomato in this particular case). Compared to the data we have obtained, much lower concentration of Pb, Cd and As (0.066, 0.011 and 0.026 mg/kg in tomato and 0.055, 0.005 and 0.013 in cabbage, respectively) have been reported by Chen et al. [45] from Xiamen, China.

The levels Zn, Cu, Fe and Mn metals were found to be 24.50, 16.57, 85.10 and 27.20 mg/kg, respectively, in tomato sample, while the corresponding values in cabbage sample were 23.53, 9.42, 490.46 and 302.23 mg/kg, respectively. The levels of Cu, Fe and Mn in cabbage and tomato samples have shown significant difference statistically at 95% probability level ($p < 0.05$), while the levels of Zn have not shown statistical difference. It has been found out that cabbage have accumulated Fe and Mn metals to significant amount compared with tomato. Cr levels on the other hand, were found to be 1.49 mg/kg in tomato sample and 4.63 mg/kg in cabbage sample. The levels of Cr obtained in this study were found to exceed allowable limit value reported by Mahmood and Malik [54] and Shaheen et al. [55]. The level of Hg in tomato sample from Mojo area farmland was 3.43 mg/kg dry weight, while the corresponding value for cabbage sample was 4.23 mg/kg dry weight of the vegetable. It can be clearly seen that both vegetable samples considered in this study are highly loaded with Hg residues and possibly imposing significant health risk to the population consuming the vegetables. The levels of Hg obtained in this study was observed to dangerously exceed the maximum limit value in vegetables as reported by Li et al., [53] and Liu et al. [73].

The Ni and Co concentrations in tomato sample were found to be 1.86 and 0.63 mg/kg, respectively, and the corresponding values in cabbage sample were 4.13 and 1.86 mg/kg, respectively. The levels of both Ni and Co were found to differ significantly at 95% probability level ($p < 0.05$) in each tomato and cabbage samples. The overall levels of heavy metals accumulation in tomato sample has followed the order of Fe > Mn > Zn > Cu > Pb > Hg > As > Ni > Cr > Co > Cd, while the order in cabbage sample followed the order of Fe > Mn > Zn > Cu > Pb > As > Cr > Hg > Ni > Co > Cd. It worth mentioning here is that, the leafy vegetable (cabbage) have accumulated heavy metals to the greater extent compared with

tomato sample. Similar research findings have been reported [11,74], in which the leafy vegetables generally accumulate heavy metals to greater extent compared with non-leafy vegetables. The high levels of heavy metals in the vegetable samples from the study area most likely attributed to anthropogenic activities in the area including the use of chemical fertilizers and/or the release of untreated solid and/or liquid wastes from the industries situated in the area.

**Bioconcentration factor (BCF).** The passage and deposit of heavy metal from soil to edible part of plants act as the main route to the entry of potentially toxic metals into the food chain [44,75]. The rates of transfer and accumulation of the heavy metals to plants vary depending upon certain factors including types of plant species, amount and types of heavy metals, physicochemical characteristics of the soil itself and other factors [44]. In respective to this, we have evaluated the transferability of heavy metals from soil to the plant species we have considered in this study (cabbage and tomato). The data for bioconcentration factor (BFC) of the heavy metals analyzed has been given in Table 5. From the data in Table 5, it can be seen that the transfer factors were increased in the order of Fe < Mn < Cr < Co < Ni < As < Pb < Cd < Zn < Hg < Cu for tomato sample. This is a clear indication that the bioaccumulation factors of copper and mercury in tomato sample were higher compared with other metals. The corresponding order for cabbage sample has followed: Fe < Co < Cr < Ni < Mn < Pb < As < Zn < Cd < Cu < Hg.

The result of this study has revealed that cabbage accumulates mercury to larger extent compared with other metals. On individual metal base, Cu was observed to accumulate in tomato to greater extent than it does in cabbage with BCF = 0.627. On the other hand, mercury (Hg) was observed to accumulate in both tomato and cabbage samples very comparably with BCF values of 0.548 and 0.579 for tomato and cabbage, respectively (Table 5). Even though the BCF values obtained in this study are all < 1, it has been observed that the leafy vegetable, cabbage, has accumulated heavy metals to greater extent compared with the fruity vegetable, tomato. The BCF values obtained for each vegetable were subjected to statistical test (one-way ANOVA test) to evaluate the presence or absence of statistical difference among cabbage and tomato. The results have revealed that the BCF values of the metals analyzed, except Zn and Hg, were found to differ significantly at 95% probability level ($p < 0.05$).

**Health risk assessments.** *Estimated Daily Intake (EDI).* The estimated daily intake (EDI) of the metals considered in this study by adult population were estimated based on the mean concentration of each metals in each food and the respective consumption rate of the vegetables as described under methods section using Eq (2) and the data including the maximum

**Table 5. Bioconcentration factor (BFC) of heavy metals analyzed for cabbage and tomato samples.**

| Metals | Bioconcentration factor (BCF) | |
|---|---|---|
| | Tomato | Cabbage |
| As | 0.079 | 0.238 |
| Pb | 0.096 | 0.211 |
| Cd | 0.106 | 0.328 |
| Zn | 0.248 | 0.251 |
| Cu | 0.627 | 0.369 |
| Fe | 0.002 | 0.012 |
| Mn | 0.015 | 0.178 |
| Cr | 0.041 | 0.129 |
| Hg | 0.548 | 0.579 |
| Ni | 0.052 | 0.135 |
| Co | 0.042 | 0.125 |

**Table 6. Estimated daily intake (mg/day/kg body weight) of toxic metals for adult population due to the consumption of contaminated vegetables in Mojo area, central Ethiopia.**

| Metals | EDI Values (mg/day/kg body weight) | | Total EDI through consumption of both tomato and cabbage | Maximum Tolerable Daily Intake (MTDI) (mg/day) |
|---|---|---|---|---|
| | Tomato | Cabbage | | |
| As | 6.06E-04 | 1.80E-03 | 2.40E-03 | 0.13[a] |
| Pb | 1.14E-03 | 2.37E-03 | 3.51E-03 | 0.21[a] |
| Cd | 1.76E-04 | 4.90E-04 | 6.65E-04 | 0.02–0.07[abc] |
| Zn | 7.69E-03 | 7.38E-03 | 1.51E-02 | 60–65[ab] |
| Cu | 5.11E-03 | 2.96E-03 | 8.06E-03 | 2.5–3[bc] |
| Fe | 2.67E-02 | 1.54E-01 | 1.81E-01 | 15[c] |
| Mn | 8.54E-03 | 9.49E-02 | 1.03E-01 | 2–5[ac] |
| Cr | 4.68E-04 | 1.45E-03 | 1.92E-03 | 0.035–0.2[ac] |
| Hg | 1.08E-03 | 1.33E-03 | 2.40E-03 | 0.04[b] |
| Ni | 5.84E-04 | 1.30E-03 | 1.88E-03 | 0.1–0.3[ac] |
| Co | 1.98E-04 | 5.84E-04 | 7.81E-04 | 0.05[c] |
| Total | 0.052 | 0.268 | 0.321 | |

[a][55]

[b][50]

[c][76]

tolerable daily intake (MTDI) for each metal is presented in Table 6. The EDI values for As, Pd, Cd and Hg were found to be $6.06 \times 10^{-4}$, $1.14 \times 10^{-3}$, $1.76 \times 10^{-4}$ and $1.08 \times 10^{-3}$ mg/day due to the consumption of 240 g/day of tomato, respectively, while the corresponding values due to the consumption of same amount of cabbage were $1.80 \times 10^{-3}$, $2.37 \times 10^{-3}$, $4.9 \times 10^{-4}$ and $1.33 \times 10^{-3}$ mg/day, respectively. The EDI of As, PB, Cd and Hg metals obtained due to the consumption of both cabbage and tomato were observed to be less than maximum tolerable daily intake of each metals as presented by Basha et al. [76]; Shaheen et al. [55] and Zheng et al. [50] and indicated in Table 6.

Likewise, the EDI of Zn, Cu, Fe, Mn, Cr, Ni and Co due to the consumption of tomato were found to be $7.69 \times 10^{-3}$, $5.11 \times 10^{-3}$, $2.67 \times 10^{-2}$, $8.54 \times 10^{-3}$, $4.68 \times 10^{-4}$, $5.84 \times 10^{-4}$ and $1.98 \times 10^{-4}$ mg/day, respectively. The corresponding EDI values due to the consumption of cabbage were $7.38 \times 10^{-3}$, $2.96 \times 10^{-3}$, $1.54 \times 10^{-1}$, $9.49 \times 10^{-2}$, $1.45 \times 10^{-3}$, $1.30 \times 10^{-3}$ and $5.85 \times 10^{-4}$ mg/day for Zn, Cu, Fe, Mn, Cr, Ni and Co, respectively. The estimated daily intake of all metals considered in this study due to the consumption of both cabbage and tomato were found to be less than the maximum tolerable daily intake as indicated in Table 6. The estimated daily intake of individual metals as a result of consumption of tomato have followed the decreasing order of Fe > Mn > Zn > Cu > Pb > Hg > As > Ni > Cr > Co > Cd. While, the intake of heavy metals due to the consumption of cabbage have followed the same decreasing order as: Fe > Mn > Zn > Cu > Pb > As > Cr > Hg > Ni > Co > Cd.

The total EDI of all the metals of interest due to the consumption of tomato was found to be 0.052 mg/day, while the corresponding value due to the consumption of cabbage was 0.268 mg/day. Even though, the EDI of metals analyzed in this study due to the consumption of tomato were found to be less than the corresponding MTDI values, the EDI of As, Cd, Pb and Zn obtained in this study were greater than the data reported by Shaheen et al. [55] for similar vegetable (tomato).

*Target hazard quotient (THQ).* The non-carcinogenic human health risk from the consumption of vegetables contaminated by heavy metals were estimated through the calculation

of target hazard quotient (THQ) as expressed under methods section using Eq (3) and the data obtained is presented in Table 7. It is evident from the data in Table 7 that the THQs of As and Hg were found to be much greater than unity in tomato sample analyzed with THQ values of 2.019 and 3.588, respectively, indicating the potential health risk due to the consumption of tomato. The corresponding THQs for the other metals analyzed were found to be < 1. Similarly, the THQ values of As, Hg, and Co were also found to be > 1 for cabbage consumption by adult population of the study area, indicating a serious potential health risk due to its consumption.

The total THQs due to the consumption of both tomato and cabbage (GTHQ) were > 1 for As, Pb, Hg and Co with GTHQ values of 8.014, 1.003, 8.014 and 2.605, respectively. This clearly suggests that the adult population in Mojo and surrounding areas (as Mojo area is the major vegetable growing area) are endangered with an alarmingly significant potential health risk by the intake of all the metals in general and As, Pb, Hg and Co (as a single metal) and/or cumulative metal contents in particular from the vegetables (tomato and cabbage) being produced and consumed in the area.

*Hazard Index (HI).* The hazard index, which considers the cumulative effect of the ingestion of various potentially hazardous metals (elements) from the consumption of different vegetables have also been computed and the data is indicated in Table 7. The HI (the sum of individual metals THQ for each vegetable (tomato and cabbage) were found alarmingly greater than unity, with HI = 7.205 and 15.078, due to the consumption of tomato and cabbage, respectively. By comparison of every THQs of metals due to the consumption of each tomato and cabbage, it can be seen that 50% of the HI due to the consumption of tomato is accounted by Hg, followed by As, Co and Pb with 28%, 9% and 5%, respectively, while the rest metals cumulatively account for only 8% (Fig 2A). Likewise, the highest contribution to HI values due to the consumption of cabbage was accounted by As (40%) followed by Hg, Co, Pb and Mn which accounts for 29, 13, 5 and 5%, respectively (Fig 2B).

Generally, about 67.7% contribution to the health index (HI) was accounted by cabbage consumption, while tomato consumption accounted for only 32.3% as can be seen from Table 7. The main contributors to the total health index (HI) were found to be As and Hg, in which both has contributed about 36% each as can be seen from Fig 3 followed by Co (12%) and Pb (5%).

It is worth mentioning here is that the present study appraised the EDI, THQ and HI values based on the estimated daily consumption of vegetables, which was 240 gram per day for both cabbage and tomato and hence it is likely that the values of EDI and THQ obtained could be overestimated and that could have possibly impacted the HI values as well. At the same time, it should also be noted that the present study had only considered cabbage and tomato for the estimation of possible noncarcinogenic and carcinogenic health risks of the population in Mojo and its surrounding. Hence, the result of this study took into account part but not the total risk to the population in the study area and as a result the potential health risks to the local population due to the exposure to heavy metals through the consumption of vegetables might be underestimated.

*The target cancer risk (TCR).* The target cancer risk (TCR) due to the exposure to heavy metals such as As, Pb, Cd, Cr and Ni through the consumption of contaminated vegetables (cabbage and tomato) were estimated by employing EDI data and oral cancer slope factor (CSPo) (mg/kg/day) as indicated under methods section and using Eqs (5) and (6). The target cancer risk due to the exposure to As, Pb, Cd, Cr and Ni through the consumption of contaminated cabbage and tomato are presented in Table 7. From the data in Table 7, it can be seen that the TCR of As due to the consumption of tomato and cabbage were 9.10 x $10^{-4}$ and 2.70 x $10^{-3}$, respectively, indicating the high risk of exposure to cancer due to the consumption of

**Table 7. THQ to heavy metals due to the consumption of contaminated vegetables (tomato and cabbage) for adults in Mojo area, central Ethiopia.**

| Metals | Target Hazard Quotient (THQ)[c] | | GTHQ[a] | Target cancer Risk (TCR)[d] | |
|---|---|---|---|---|---|
| | Tomato | Cabbage | | Tomato | Cabbage |
| As | **2.019** | **5.994** | 8.014 | **9.10E-04** | **2.70E-03** |
| Pb | 0.326 | 0.678 | 1.003 | 9.70E-06 | 2.02E-05 |
| Cd | 0.176 | 0.490 | 0.665 | 6.69E-05 | **1.86E-04** |
| Zn | 0.026 | 0.025 | 0.050 | - | - |
| Cu | 0.128 | 0.074 | 0.202 | - | - |
| Fe | 0.038 | 0.220 | 0.258 | - | - |
| Mn | 0.061 | 0.678 | 0.739 | - | - |
| Cr | 0.156 | 0.484 | 0.640 | **2.34E-04** | **7.28E-04** |
| Hg | **3.588** | **4.425** | 8.014 | - | - |
| Ni | 0.029 | 0.065 | 0.094 | **9.94E-04** | **2.21E-03** |
| Co | 0.659 | **1.946** | 2.605 | - | - |
| HI[b] | 7.205 | 15.078 | **22.283** | - | - |

[a] GTHQ is the sum of individual metals THQ for every vegetable

[b] HI is Hazard Index

[c] values indicated in bold have shown THQ > 1

[d] values indicated in bold have exceeded the upper limit $(1 \times 10^{-4})$ for acceptable risk of developing cancer

these vegetables from the area as the values were exceeded the maximum threshold value of $1 \times 10^{-4}$ [11,45,48,50,52,53,73,77–79]. The corresponding TCR of Pb were 9.70 x 10–6 and 2.02 x 10–5, respectively due to the consumption tomato and cabbage. The TCR values for Pb were observed to be less the maximum threshold value and hence indicating no cancer risk from Pb to the adult population in the area through consumption of both tomato and cabbage. In compression with the literature reports, the TCR value for As $(1.9 \times 10^{-8})$ reported by Shaheen et al. [55] was found to be much less that the value $(9.10 \times 10^{-4})$ we have reported in this study. Similarly, Antoine and co-workers [52] have reported a TCR value of $7.61 \times 10^{-6}$ for As due to the consumption of tomato, which is much less the data we have reported in this study.

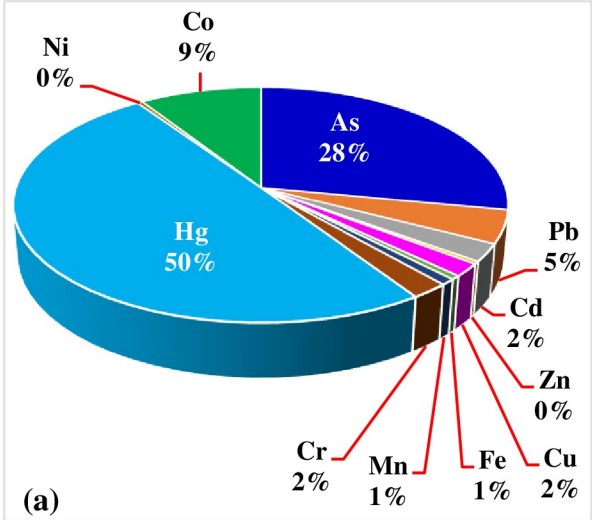 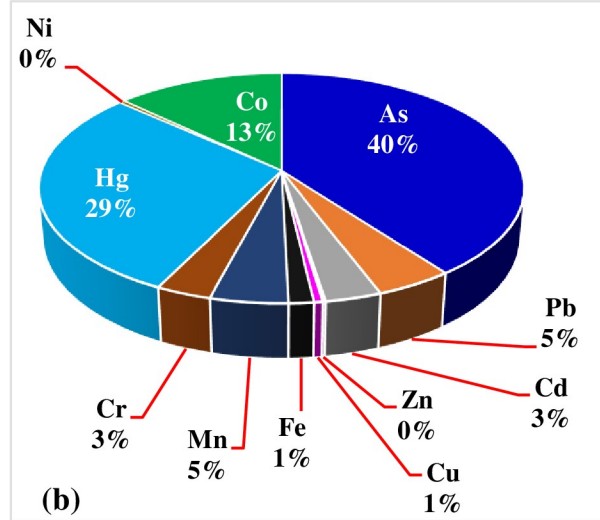

**Fig 2.** The average contribution of heavy metals to the HI due to the consumption of tomato (a) and cabbage (b).

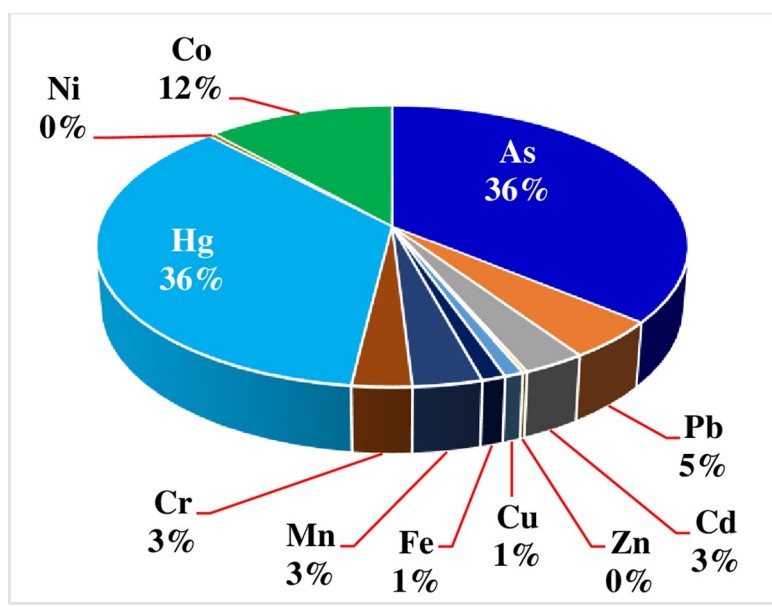

**Fig 3. Percentage contribution of each metals to the total HI.**

The cancer risk from Cd to the adult population through the consumption of cabbage observed to be positive as the TCR of Cd (1.86 x 10–4) was observed to exceed the threshold value indicated. Cr and Ni are both regarded as ultimately causing cancer risk to the adult population in Mojo area and its surroundings through the consumption of both tomato and cabbage vegetables as the TCR values for both Cr and Ni exceeded the threshold value (Table 7). Generally, 4 out of 5 (80%) of the heavy metals in cabbage sample for which the TCR values estimated were found to instigate cancer risk to the adult population in the area of study. However, 3 out of 5 (60%) of the heavy metals in tomato samples for which the TCR values estimated were found to instigate cancer risk to the adult population in Mojo area and its surroundings.

## Conclusion

The result of our study has revealed that the levels of As, Pb, Cd, Cu, Hg and Co were found to exceed the recommended values in agricultural soil. Similarly, As, Pb, Cd, Cr and Hg were found in an alarmingly higher concentration in both tomato and cabbage samples analyzed. The estimated daily intake of toxic metals due to the consumption of both cabbage and tomato were found fairly below the maximum tolerable daily intake proposed for each metal. However, from the human health implication point of view, it was found out that the THQ to the heavy metals due to the consumption of tomato were > 1 for As (2.019) and Hg (3.588). Likewise, THQ were also > 1 for As (5.994), Hg (4.425) and Co (1.946) due to the consumption of cabbage. The TDHQ to heavy metals due to the consumption of both cabbage and tomato were > 1 for As (8.014), Pb (1.003), Hg (8.014) and Co (2.605) indicating about 72% of TDHQ were accounted by As and Hg. The combined noncarcinogenic effects of multiple metals as estimated by the HI were found to exceed 1 due to the consumption of each tomato (HI = 7.025) and cabbage (HI = 15.078), indicating about 67.7% of the effect is accounted for the consumption of cabbage alone. The carcinogenic effect analysis has revealed that the total cancer risk (TCR) from As, Cr and Ni due to the consumption of tomato were found to be > $10^{-4}$ (the maximum threshold value). Similarly, the TCR values for As, Cd, Cr and Ni

due to the consumption of cabbage were also exceeded the maximum threshold value of $10^{-4}$. This generally suggests the presence of potential cancer risk to the population from As, Cd, Cr and Ni due to the consumption of both tomato and cabbage being cultivated in Mojo area and its surrounding in central Ethiopia.

## Supporting information

**S1 Table. Optimal conditions achieved for soil and vegetables samples digestion procedures.**
(PDF)

**S2 Table. Instrumental operating condition for the analysis of metal in soil and vegetable samples.**
(PDF)

**S3 Table. Method detection limits (MDL) and limit of quantification (LOQ) for vegetable and soil samples analysis.**
(PDF)

**S4 Table. Percentage recovery data of the method used for soil sample digestion (M±SD, n = 3).**
(PDF)

**S5 Table. Percentage recovery values of the method used for tomato digestion (M±SD, n = 3).**
(PDF)

**S6 Table. Percentage recovery values of the method used for cabbage digestion (M±SD, n = 3).**
(PDF)

## Acknowledgments

We would like to thank Ambo University and Agricultural and Nutritional Research Laboratory of Ethiopian Institute of Agricultural Research located in Addis Ababa for the laboratory facilities.

## Author Contributions

**Conceptualization:** Leta Danno Bayissa.

**Data curation:** Hailu Reta Gebeyehu.

**Formal analysis:** Hailu Reta Gebeyehu.

**Investigation:** Hailu Reta Gebeyehu.

**Methodology:** Leta Danno Bayissa.

**Supervision:** Leta Danno Bayissa.

**Validation:** Leta Danno Bayissa.

**Writing – original draft:** Leta Danno Bayissa.

**Writing – review & editing:** Leta Danno Bayissa.

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
