## [Decision Letter · Decision Letter 0]

25 Nov 2019

PONE-D-19-28586

Levels of heavy metals in soil and selected vegetables and associated health risks to the population in Mojo area, central Ethiopia

PLOS ONE

Dear Dr Bayissa,

Thank you for submitting your manuscript to PLOS ONE. After careful consideration, we feel that it has merit but does not fully meet PLOS ONE’s publication criteria as it currently stands. Therefore, we invite you to submit a revised version of the manuscript that addresses the points raised during the review process.

ACADEMIC EDITOR: 

Dear Authors,

Reviewers have now commented on your paper. For your guidance, reviewers' comments are appended below. You will see that they are advising that you revise your manuscript. If you are prepared to undertake the work required, I would be pleased to reconsider my decision. Please give a careful consideration to the criticisms raised. All of them should be taken into account in a revised version of your manuscript.

I thank you for giving us the possibility of considering your work and looking forward to receive your revised manuscript.

Best regards,

Amit Bhatnagar, PhD

We would appreciate receiving your revised manuscript by Jan 09 2020 11:59PM. To enhance the reproducibility of your results, we recommend that if applicable you deposit your laboratory protocols in protocols.io, where a protocol can be assigned its own identifier (DOI) such that it can be cited independently in the future. For instructions see: http://journals.plos.org/plosone/s/submission-guidelines#loc-laboratory-protocols

We look forward to receiving your revised manuscript.

Kind regards,

Amit Bhatnagar, Ph. D.

Academic Editor

PLOS ONE

Journal Requirements:

3. In your Methods section, please provide additional information regarding the permits you obtained for the work. Please ensure you have included the full name of the authority that approved the field site access and, if no permits were required, a brief statement explaining why. Please also clarify if the vegetables analysed in your study were collected from a shop or a farm and provide their name and location.

This research project was financially supported by Ethiopian Institute of Agricultural Research (EIAR).

The project was financially supported by Ambo University. The funders had no role in study design, data collection and analysis, decision to publish, or preparation of the manuscript.

Additional Editor Comments:

Dear Authors,

Reviewers have now commented on your paper. For your guidance, reviewers' comments are appended below. You will see that they are advising that you revise your manuscript. If you are prepared to undertake the work required, I would be pleased to reconsider my decision. Please give a careful consideration to the criticisms raised. All of them should be taken into account in a revised version of your manuscript.

I thank you for giving us the possibility of considering your work and looking forward to receive your revised manuscript.

Best regards,

Amit Bhatnagar, PhD

Reviewers' comments:

Reviewer's Responses to Questions

**Comments to the Author**

1. Is the manuscript technically sound, and do the data support the conclusions?

Reviewer #1: Yes

Reviewer #2: Yes

2. Has the statistical analysis been performed appropriately and rigorously? 

Reviewer #1: Yes

Reviewer #2: Yes

3. Have the authors made all data underlying the findings in their manuscript fully available?

Reviewer #1: Yes

Reviewer #2: Yes

4. Is the manuscript presented in an intelligible fashion and written in standard English?

Reviewer #1: Yes

Reviewer #2: Yes

5. Review Comments to the Author

Reviewer #1: The authors need to recheck references to ensure they are in conformity with the Journal format. Also, check spaces between paras. The conclusion need to be shortened to one para and should not duplicate results.

Reviewer #2: The tittle of manuscript is long; it can be changed to: “Levels of heavy metals in soil and vegetables and associated health risks in Mojo, Ethiopia”

In abstract section line 14-16, delete “Properly optimized digestion techniques were employed to digest the soil and vegetable samples after appropriate validation through recovery study which were ranged from 90 to 117.46 % with relative standard deviations (RSD) < 11.6 %.” Fro this section.

In introduction section please add more information about soil pollution from heavy metals and add this papers to this section:

https://doi.org/10.1007/s13762-017-1327-x

https://doi.org/10.1080/10807039.2018.1460798

Page 22, line 56, please add this paper to references as one of the similar work in the world and compare your obtained results with similar work:

https://doi.org/10.1016/j.dib.2018.10.114

Page 5, lin 115, please use full name of chemicals for “NaAc” and “NH4Ac”

Please add more information about “Optimization of Digestion Procedures for Soil and Vegetable Samples”.

Why didn’t used calculation of EDI, THQ and TCR for children in table 2 ?

Conclusion is too long; the authors need to improve their conclusion. Include information based on major finding from the research.

6. PLOS authors have the option to publish the peer review history of their article (what does this mean?). If published, this will include your full peer review and any attached files.

Reviewer #1: Yes: David Sylvester Kacholi

Reviewer #2: No

---

## [Author Response · Author response to Decision Letter 0]

4 Dec 2019

Response to Reviewers Comment

Reviewer #1

The comments forwarded are very constructive and helped us to enrich our paper.

-We have carefully checked the references and made necessary corrections in conformity with the journal format. 

-The space between the paragraphs have also been checked and any irregularities were corrected.

-We have shortened the conclusion as per the suggestion.

Reviewer #2

We really appreciate the reviewer for the genuine and constructive comments. The comment given are very helpful and therefore, all the suggestions have been given due attention and corrections have been made accordingly.

-The title has been shortened as suggested

-The statements suggested to be removed from the abstract section have been removed accordingly.

-We have included one of the reference suggested by the reviewer while the others are some what similar with the already cited references.

-Line 115 >> we have made the suggested corrections

-The optimization of the digestion procedures have been explained and based on the suggestion given we have tried to elaborate it more. The detain procedures and the results obtained have also been explained under Results and Discussion section as well.

-In this particular study, we have focused on adults as adults are the prominent consumer of the indicated vegetables. In principle it is necessary and again possible to calculate the EDI, THQ and TCR for children as well. However, for the sake of focus only, we have concentrated on adults.

-We have shortened the conclusion part as per the suggestion.

---

## [Decision Letter · Decision Letter 1]

24 Dec 2019

PONE-D-19-28586R1

Levels of heavy metals in soil and vegetables and associated health risks in Mojo area, Ethiopia

PLOS ONE

Dear Dr Bayissa,

Thank you for submitting your manuscript to PLOS ONE. After careful consideration, we feel that it has merit but does not fully meet PLOS ONE’s publication criteria as it currently stands. Therefore, we invite you to submit a revised version of the manuscript that addresses the points raised during the review process.

ACADEMIC EDITOR: 

Dear Authors,

After 2nd round of review, one reviewer suggested that your manuscript can be accepted after minor revisions. Please consider these comments and address these in your revised manuscript.

Best regards,

Amit Bhatnagar

We would appreciate receiving your revised manuscript by Feb 07 2020 11:59PM. To enhance the reproducibility of your results, we recommend that if applicable you deposit your laboratory protocols in protocols.io, where a protocol can be assigned its own identifier (DOI) such that it can be cited independently in the future. For instructions see: http://journals.plos.org/plosone/s/submission-guidelines#loc-laboratory-protocols

We look forward to receiving your revised manuscript.

Kind regards,

Amit Bhatnagar, Ph. D.

Academic Editor

PLOS ONE

Reviewers' comments:

Reviewer's Responses to Questions

**Comments to the Author**

1. If the authors have adequately addressed your comments raised in a previous round of review and you feel that this manuscript is now acceptable for publication, you may indicate that here to bypass the “Comments to the Author” section, enter your conflict of interest statement in the “Confidential to Editor” section, and submit your "Accept" recommendation.

Reviewer #2: (No Response)

2. Is the manuscript technically sound, and do the data support the conclusions?

Reviewer #2: No

3. Has the statistical analysis been performed appropriately and rigorously? 

Reviewer #2: Yes

4. Have the authors made all data underlying the findings in their manuscript fully available?

Reviewer #2: (No Response)

5. Is the manuscript presented in an intelligible fashion and written in standard English?

Reviewer #2: Yes

6. Review Comments to the Author

Reviewer #2: Authors need to improve the introduction and add the soil pollution and add this paper to references list: https://doi.org/10.1007/s13762-017-1327-x

Conclusion section is too long yet, need to improvement of conclusion as major finding of work.

7. PLOS authors have the option to publish the peer review history of their article (what does this mean?). If published, this will include your full peer review and any attached files.

Reviewer #2: No

---

## [Author Response · Author response to Decision Letter 1]

24 Dec 2019

Response to Reviewers

Reviewer #2

1. We have revisited the introduction part and included the recommended reference.

2. We checked once again the conclusion section and shortened it as much as possible as per the suggestion from the respected reviewer.

---

## [Decision Letter · Decision Letter 2]

2 Jan 2020

Levels of heavy metals in soil and vegetables and associated health risks in Mojo area, Ethiopia

PONE-D-19-28586R2

Dear Dr. Bayissa,

We are pleased to inform you that your manuscript has been judged scientifically suitable for publication and will be formally accepted for publication once it complies with all outstanding technical requirements.

With kind regards,

Amit Bhatnagar, Ph. D.

Academic Editor

PLOS ONE

Additional Editor Comments (optional):

Reviewers' comments:

Reviewer's Responses to Questions

**Comments to the Author**

1. If the authors have adequately addressed your comments raised in a previous round of review and you feel that this manuscript is now acceptable for publication, you may indicate that here to bypass the “Comments to the Author” section, enter your conflict of interest statement in the “Confidential to Editor” section, and submit your "Accept" recommendation.

Reviewer #2: All comments have been addressed

2. Is the manuscript technically sound, and do the data support the conclusions?

Reviewer #2: Yes

3. Has the statistical analysis been performed appropriately and rigorously? 

Reviewer #2: Yes

4. Have the authors made all data underlying the findings in their manuscript fully available?

Reviewer #2: Yes

5. Is the manuscript presented in an intelligible fashion and written in standard English?

Reviewer #2: Yes

6. Review Comments to the Author

Reviewer #2: (No Response)

7. PLOS authors have the option to publish the peer review history of their article (what does this mean?). If published, this will include your full peer review and any attached files.

Reviewer #2: No

---

## [Editor Report · Acceptance letter]

6 Jan 2020

PONE-D-19-28586R2 

Levels of heavy metals in soil and vegetables and associated health risks in Mojo area, Ethiopia 

Dear Dr. Bayissa:

I am pleased to inform you that your manuscript has been deemed suitable for publication in PLOS ONE. Congratulations! Your manuscript is now with our production department. 

With kind regards,

on behalf of

Dr. Amit Bhatnagar 

Academic Editor

PLOS ONE